# Serum creatinine and estimated glomerular filtration rate (eGFR) in early pregnancy and changes during the pregnancy

**Suneth Buddhika Agampodi**[1]*, **Thilini Chanchala Agampodi**[1], **Gayani Shashikala Amarasinghe**[1], **Janith Niwanthaka Warnasekara**[1], **Ayesh Umeshana Hettiarachchi**[1], **Imasha Upulini Jayasinghe**[1], **Iresha Sandamali Koralegedara**[2], **Parami Abeyrathna**[3], **Shalka Srimantha**[1], **Farika Nirmani de Silva**[1], **Sajaan Praveena Gunarathne**[1], **Nuwan Darshana Wickramasinghe**[1]

1 Department of Community Medicine, Faculty of Medicine and Allied Sciences, Rajarata University of Sri Lanka, Mihintale, Sri Lanka, 2 Department of Anatomy, Faculty of Medicine and Allied Sciences, Rajarata University of Sri Lanka, Mihintale, Sri Lanka, 3 Department of Family Medicine, Faculty of Medicine and Allied Sciences, Rajarata University of Sri Lanka, Mihintale, Sri Lanka

* sunethagampodi@yahoo.com

**Data Availability Statement:** All data used in this manuscript is uploaded as supplementary file.

## Abstract

Renal functions in pregnancy undergo rapid changes, and the thresholds for normal values are a major research gap and are still debatable. The lack of prospective population-based studies with early pregnancy recruitment hampered the decision-making process on the best thresholds to be used in clinical practice. We present the serum creatinine (sCr) and sCr-based estimated glomerular filtration rates (eGFR) in early pregnancy with changes over the gestational period in a large prospective, community-based cohort, the Rajarata Pregnancy Cohort (RaPCo). We carried out a community-based prospective cohort study with 2,259 healthy pregnant women with a gestation period of less than 13 weeks and without pre-existing medical conditions. Gestational period-specific sCr and sCr-based eGFR were calculated for different age strata, and the participants were followed up until the second trimester. Renal functions of pregnant women were compared with 2.012 nonpregnant women from the same geographical area. The mean (SD) sCr of the 2,012 nonpregnant women was 62.8(12.4) μmol/L, with the 97.5th percentile of 89.0 μmol/L. Among the pregnant women, mean (SD) sCr was 55.1(8.3), 52.7(8.1), 51.1(9.1), 47.1(7.2), and 49.3 (9.9), while the 97.5th percentile for sCr was 72.4, 69.1, 70.0, 63.6, and 66.0 μmol/L respectively during the 4–7, 8–9, 10–12, 24–27 and 28–30 weeks of gestation. The average sCr value was 84.7% and 76.4% of the nonpregnant group, respectively, in the first and second trimesters. The mean eGFR was 123.4 (10.7) mL/min/1.73 m$^2$ in the first trimester and increased up to 129.5 mL/min/1.73 m$^2$ in the 24th week of gestation. The analysis of cohort data confirmed a significant reduction in sCr with advancing pregnancy (p<0 .001). This study provides thresholds for renal functions in pregnancy to be used in clinical practice. Clinical validation of the proposed thresholds needs to be evaluated with pregnancy and newborn outcomes.

**Funding:** The original cohort study was supported by the Accelerating Higher Education Expansion and Development (AHEAD) Operation of the Ministry of Higher Education, Sri Lanka funded by the World Bank [grant number DOR STEM HEMS [6026-LK/8743-LK] to TCA]. The funding agency has no role in the design of the study and collection, analysis, interpretation of data and in writing the manuscript.

**Competing interests:** The authors have declared that no competing interests exist.

## Introduction

Owing to the alteration of the renin-angiotensin-aldosterone system (RAAS) and other maternal hormonal changes, systemic vascular resistance decreases during pregnancy, leading to lower blood pressure and an increase in renal plasma flow [1]. Studies using inulin, paminohippurate clearances [2] and 24-hour creatinine clearance [3] suggest that with augmented blood flow, renal vasodilatation and volume expansion up to 70%, a progressive increase in glomerular filtration occurs during pregnancy. Due to the rapid and dynamic changes in renal physiology during pregnancy, assessing renal functions and deciding on thresholds for normal and abnormal values is a major challenge in clinical practice.

Assessing renal functions in routine obstetric practise is a challenge. Both cystatin C- and serum creatinine (sCr)-based equations have been shown to systematically underestimate the glomerular filtration rate (GFR) in the pregnancy [4–7]. Thus, 24-hour urine collection remains the standard method for estimating GFR. At the same time, sCr is used in clinical settings as a more feasible test for assessing renal functions in routine practice [8]. Nevertheless, both sCr- and sCr-based eGFR were shown to predict adverse outcomes in pregnancy [9–11], although the latter was proven to be an inaccurate estimate of renal functions in pregnancy [12].

A recent study using electronic data from 243,534 pregnancies showed that sCr rapidly decreases from 60 μmol/L prepregnancy to approximately 47 μmol/L at 16–32 weeks [13]. This observation is consistent with the findings of two recent systematic reviews [14, 15]. One review [14] included 49 studies with 4,421 serum creatine measurements. The authors proposed 85%, 80%, and 86% of the nonpregnant sCr upper limit in sequential trimesters as the standards for deciding "abnormal" values. Another systematic review [15] included 29 studies in the analysis and showed that sCr reduction was most prominent at 15–21 weeks of gestation, with a 23.2% reduction, slightly more than the percentage estimated in the previous systematic review. Both systematic reviews discussed several limitations in published literature, including small samples size, heterogeneous nature of studies, retrospective/ secondary data use and use of sCr values, which were requested based on clinical grounds.

Against the backdrop of these critical evidence gaps, the present study was designed to assess sCr and sCr based eGFR of pregnant women using a population-based prospective cohort design with comparable reference data drawn from the same reference population without sampling bias.

## Methods

### Ethics statement

Written informed consent was obtained from all pregnant women before the enrolment. For the minors, consent was obtained from the parent/ legal guardian. All procedures performed were in accordance with the ethical standards of the institution and the 1964 Helsinki declaration. Ethical clearance for the RaPCo study was obtained from the ethics review committee of the Faculty of Medicine and Allied Sciences, Rajarata University of Sri Lanka (ERC/2019/07).

This study was a component of the Rajarata Pregnancy Cohort (RaPCo) [16]. The study was performed in Anuradhapura, the largest district (geographically) in Sri Lanka. All pregnant women newly registered from July to September 2019 and residing in Anuradhapura were invited to participate in RaPCo. It recruited more than 90% of newly registered pregnant women in the district.

Out of the 3,374 pregnant women recruited for the RaPCo study, all pregnant women over 18 years of age with a period of gestation (PoG) less than 12 weeks at recruitment were

included in the present study. PoG was confirmed retrospectively after the dating ultrasound scan. The exclusion criteria included pregnant women with uncertain dates; a history of physician-diagnosed renal diseases, hypertension, diabetes mellitus, ischemic heart diseases, hyperlipidemia, autoimmune diseases, and thyroid dysfunctions; pregnant women with any renal disorders, hypertensive disorders and hyperglycemic conditions in previous pregnancies and multiple pregnancies. At the baseline assessment, a 75 g OGTT was performed, and all pregnant women with fasting plasma glucose greater than 126 mg/dL and 2-hr plasma glucose greater than 200 mg/dL were excluded. Pregnant women with systolic blood pressure greater than 140 mmHg and/or diastolic blood pressure greater than 90 mmHg at the first visit (screened using Omron OMRON HEM-7320) were also excluded. Those who had urine protein (1+ or above) and haematuria (>4 RBC) in the dipstick test during the baseline assessment were also removed from this study. A follow-up assessment was performed towards the end of the second trimester. Study participants were invited to participate in the follow-up clinic at approximately 24–28 weeks of gestation. Only those who attended the clinics at 24–30 weeks (some participants were a bit late in clinic attendance) and had baseline and follow-up measurements were included in the follow-up (repeated measure) analysis.

A sample of venous blood was collected in a plain tube by a qualified nursing officer. All collected samples were stored at -80˚C for further analysis. Serum creatinine was assessed using a creatinine-sarcosine oxidase method (CREA-S) assay kit (a isotope dilution mass spectometry traceable creatinine assay) in a fully automated Mindray BS-240 clinical chemistry analyzer. For the estimation of eGFR, the Chronic Kidney Disease Epidemiology Collaboration (CKD-EPI) formula was used.

A sample of healthy non-pregnant females of reproductive age participating in a large community-based chronic kidney disease (CKD) screening program in Anuradhapura in 2015–16 were recruited as the comparison group [17]. To define the age groups (nonpregnant women) and PoG groups (pregnant women) for the analysis, a homogenous subset identification table of ANOVA was used. A one-way between-group ANOVA was conducted to explore the impact of PoG on sCr. Post hoc comparisons were made using Tukey's HSD test to compare groups. A two-way ANOVA analysis was done to explore the effect of age and PoG on sCr. Finally, a repeated-measures ANOVA was used to re-examine/confirm the finding using paired samples among whom both first-trimester and second-trimester values are available. The proposed threshold for 'abnormal' sCr was based on the 97.5th percentile.

## Results

### Renal functions of nonpregnant women of reproductive age

Data from 2,012 nonpregnant women of reproductive age were available for the comparison group. The mean (SD) age of this group was 35.9 (8.2) years. The mean (SD) sCr of the group was 62.8 (12.4) µmol/L with a 97.5th percentile of 89.0 µmol/L. The mean (SD) eGFR was 105.1 (27.9) mL/min/1.73 m$^2$ with a median of 100.8 (IQR 85.5–118.4) mL/min/1.73 m$^2$. The distribution of sCr and eGFR showed that the sCr values were fairly normally distributed (skewness of 0.182), while eGFR was skewed to the right (skewness of 1.21). However, the Kolmogorov-Smirnov statistics showed a violation of normality for both parameters (p<0.001). Age-disaggregated mean sCr and eGFR values were compared to identify homogenous subsets of age groups using one-way ANOVA. Based on the ANOVA results, two subgroups for sCr and three subgroups for eGFR were identified (Table 1). Although the mean sCr was significantly different, the 95th percentile for sCr was almost similar.

**Table 1. Distribution of serum creatinine and eGFR values by age among nonpregnant females of reproductive age in North Central Province, Sri Lanka.**

| Age (years) | N | Mean | Std. Deviation | Std. Error | 97.5th Percentile |
|---|---|---|---|---|---|
| Serum creatinine (µmol/L) | | | | | |
| <35 | 920 | 60.93 | 12.383 | 0.408 | 88.6 |
| 35 and above | 1,092 | 64.45 | 12.222 | 0.370 | 89.2 |
| eGFR (mL/min/1.73 m²) | | | | | |
| <30 | 534 | 118.7 | 30.533 | 1.321 | |
| 30–34 | 386 | 107.19 | 25.624 | 1.304 | |
| 35 and above | 1,092 | 97.72 | 24.418 | 0.739 | |

### The pregnancy cohort

Data from A total of 2,407 pregnant women were included in this analysis (S1 Data). The mean (SD) age of the study sample was 28.0 (5.4) years. The numbers of pregnant women in their first, second and third pregnancies were 698 (29.0%), 803 (33.4%) and 611 (25.4%), respectively.

The first-trimester sCr in the pregnancy cohort was reasonably normally distributed (skewness -0.004) with a mean (SD) of 53.2 (8.6) µmol/L. The 97.5th percentile for the sCr in the first trimester was 70.6 µmol/L. The mean (SD) eGFR was 123.4 (10.6) mL/min/1.73 m², with a median of 123.5 (IQR 118.1–129.4) mL/min/1.73 m².

During the 24–30 weeks of PoG (979 pregnant women), the mean SC (SD) was 47.7 (8.1) µmol/L, with a slightly skewed distribution (skewedness 1.9). The 97.5th percentile for sCr from 24⁻30 weeks was 65.1 µmol/L. The mean (SD) eGFR was 129.0 (10.0) mL/min/1.73 m², with a median of 129.2 (IQR 123.6–135.3) mL/min/1.73 m².

From the 4th-5th week of PoG, SC continued to decrease steadily until the completion of 12 weeks (Fig 1). In the 24th week, a further decline in sCr was observed, and it started to increase after the 25th week. The respective eGFR values followed the inverse pattern, with the highest value at approximately the 24th week.

Homogenous subsets of sCr values according to PoG were prepared for further analysis using one-way ANOVA (Table 2). In these groups, a one-way between-group ANOVA was

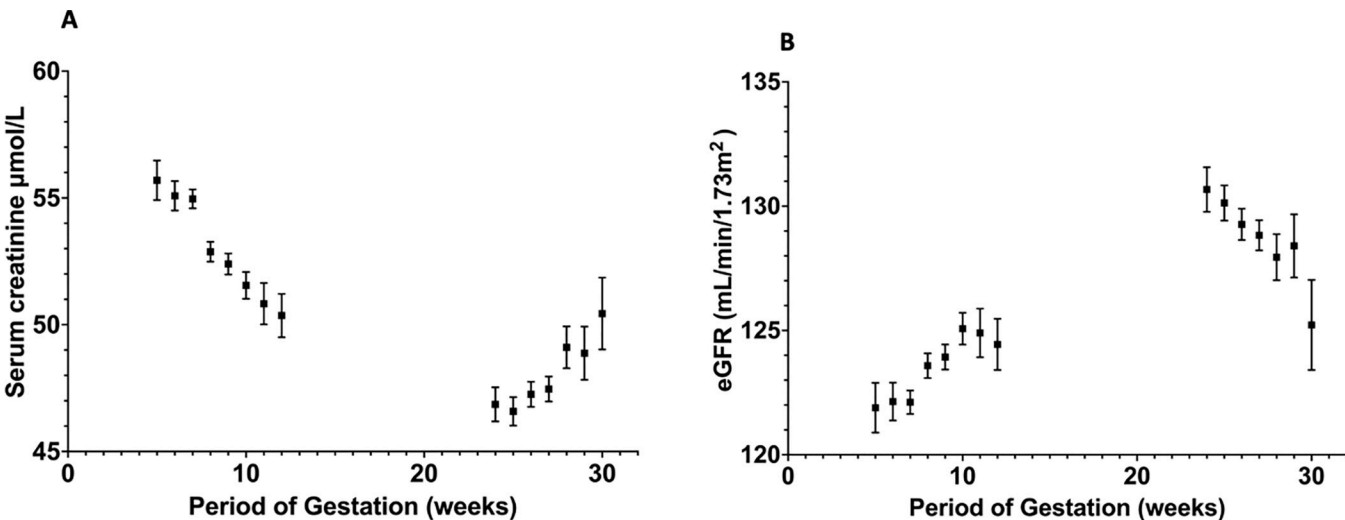

**Fig 1. Distribution of serum creatinine (A) and eGFR (B) by the period of gestation in 2407 pregnant women during the first trimester and follow-up at 24–30 weeks.**

**Table 2. Distribution of serum creatinine and eGFR by period of gestation (grouped) in 2407 pregnant women during the first trimester and follow-up in Anuradhapura, Sri Lanka.**

| PoG (Weeks) | N | Serum creatinine (µmol/L) | | | | | eGFR | |
|---|---|---|---|---|---|---|---|---|
| | | Mean | SD | 95% CI for Mean Lower-Upper | | 97.5th Percentile | Mean | SD |
| 4–7 | 794 | 55.1 | 8.3 | 54.5 | 55.7 | 72.4 | 122.1 | 10.7 |
| 8–9 | 808 | 52.7 | 8.1 | 52.1 | 53.2 | 69.1 | 123.7 | 10.1 |
| 10–12 | 515 | 51.1 | 9.1 | 50.3 | 51.9 | 70.0 | 124.9 | 10.9 |
| 24–27 | 700 | 47.1 | 7.2 | 46.6 | 47.7 | 63.6 | 129.5 | 9.2 |
| 28–30 | 279 | 49.3 | 9.9 | 48.1 | 50.5 | 66.0 | 127.6 | 11.7 |

conducted to explore the impact of PoG on sCr. There was a significant difference in sCr for the five groups [F (3,095) = 94.375, p<0.001]. The effect size calculated using eta squared was 0.105 (medium to large effect). Post hoc comparisons using Tukey's HSD test indicated that the mean sCr for each PoG group was significantly different from that of other adjacent groups.

As age showed an effect on sCr in the nonpregnant cohort, the values of the pregnancy cohort were further analyzed according to age categories (Fig 2).

After the initial descriptive analysis and subset analysis, participants were divided into two groups according to their age (Group 1: less than 35 years; Group 2: 35 years and above). Using a two-way ANOVA for age and PoG, the interaction effect between PoG and age group was found to be marginal [F(4,3086) = 2.058, p = 0.084]. There was a statistically significant main effect for PoG [F(4,3086) = 41.347, p<0.001], and the effect size was small to medium (partial eta squared = 0.051). Post hoc comparisons using Tukey's HSD test indicated that even after including age in the model, the mean sCr for each PoG group was significantly different from that of the adjacent groups. The main effect of age [F(1,3086) = 3.161, p = 0.076] was not significant.

To further assess the changes in renal functions using the cohort design, a one-way repeated-measures ANOVA was conducted. This analysis was conducted only for those who had both baseline and follow-up data at 24–27 weeks of PoG, in which the lowest sCr was observed (n = 500). Three groups were defined according to the PoG at the time of recruitment as above. The mean (SD) values of the first and second measures are presented in Table 3. There was a significant reduction in sCr with advancing pregnancy [Wilks' Lambda = 0.705, F (1,499) = 208.863, p<0.001, multivariate partial eta squared = 0.288]. This analysis showed that despite having different mean values based on the PoG, at 24–27 weeks, the PoG values were concentrated around a mean value of 47 µmol/L.

## Discussion

In this prospective cohort study, we systematically recruited a population-based sample of women with singleton pregnancies, excluding all comorbidities, to generate proper "normality data" for sCr in pregnancy. This prospective study, probably one of the largest reported so far for the first-trimester renal function assessment in pregnancy with 2407 pregnant women and with 979 follow-ups, provides the renal function thresholds in pregnancy for the South Asian region.

Different upper normal limits for sCr have been proposed without consensus for many years. The suggested values varied, with different studies reporting 72 µmol/L [18], 89 µmol/L [19], 80 µmol/L [20] and 95 µmol/L [21] as upper limits. A similar study performed recently in China also published higher upper values of 68, 66, and 68 µmol/L for the first, second, and

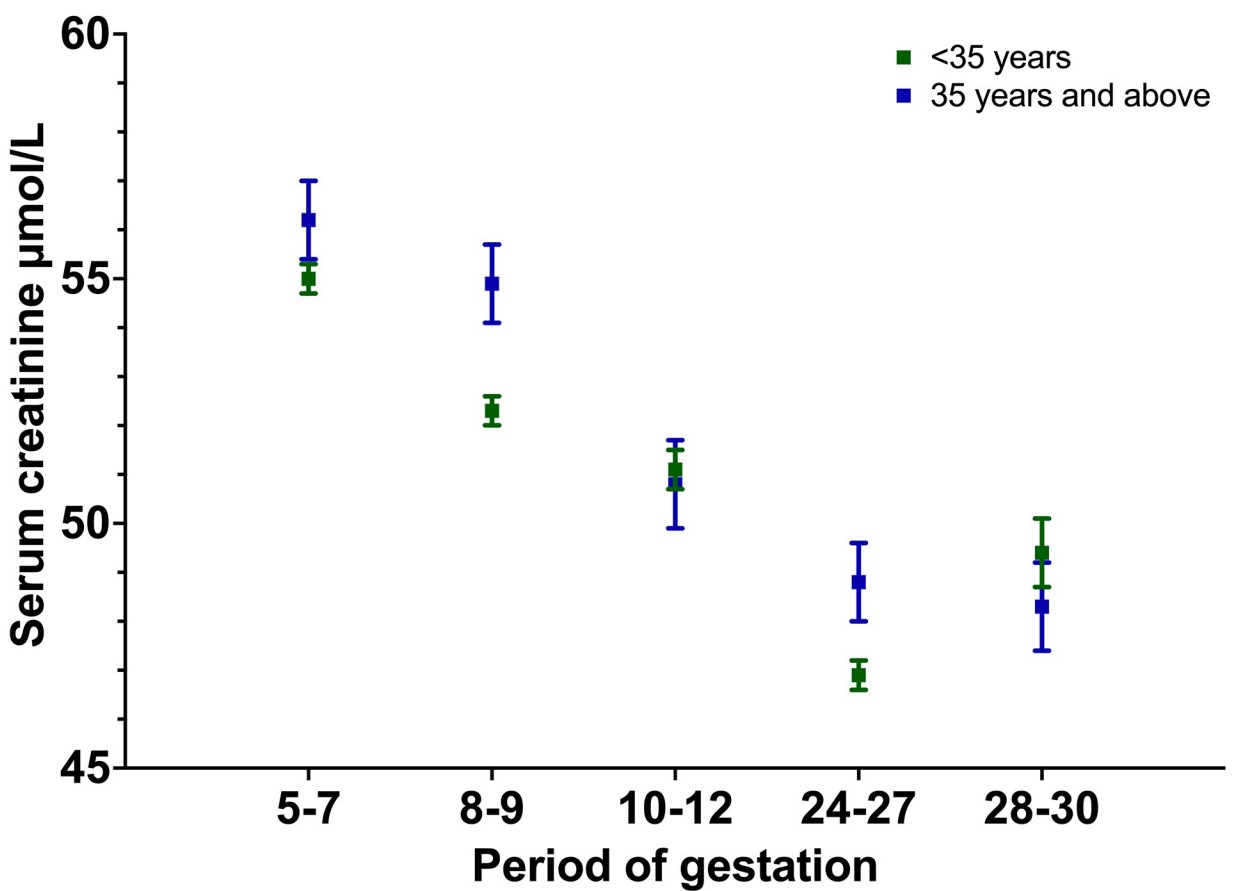

**Fig 2. Distribution of serum creatinine by period of gestation and age in 2407 first trimester pregnant women in Anuradhapura, Sri Lanka.**

third trimesters, respectively [22]. In 2019, the Renal Association comprehensively reviewed the published guidelines from the National Institute of Health and Care Excellence (NICE), UK Consensus Group on Pregnancy in Renal Disease, and Kidney Disease Outcomes Quality Initiative (KDOQI) and searched Ovid Medline (1946 to 2018) for "Clinical practice guideline on pregnancy and renal disease" [8]. This guideline used the two most recent reviews: the Canadian study [13] and the systematic review published in 2019. Compared with the 95th percentile reported in the Canadian study, the 95th percentiles observed in our study cohort in the

**Table 3. Results of paired sample (baseline and followup) serum creatinine among 500 pregnant women with follow-up measurements performed between 24–27 weeks.**

| | | Serum creatinine µmol/L | | | |
| --- | --- | --- | --- | --- | --- |
| | | 1st trimester (5–12 weeks) | | End of 2nd trimester (24-27th weeks) | |
| PoG at the first visit | N | Mean | SD | Mean | SD |
| 4–7 weeks | 148 | 54.5 | 8.5 | 47.2 | 8.1 |
| 8–9 weeks | 228 | 52.8 | 8.5 | 47.7 | 7.3 |
| 10–12 weeks | 124 | 49.6 | 9.1 | 47.2 | 7.0 |
| Total | 500 | 52.50 | 8.84 | 47.4 | 7.5 |

first and second trimesters were slightly higher. In weeks 4–7, 8–9, 10–12, 24–27 and 28–30, a previous study reported 70, 65, 61, 59 and 59 μmol/L, respectively. Compared with the systematic review, which reported 85% and 80% of prepregnancy sCr values in the first and second trimesters, we observed values of 84.7% and 76.4% compared with the nonpregnant group, respectively, showing a slightly higher decrease at the end of the second trimester. In our study, we tried to overcome the listed limitations in both studies by using a prospective design and including all "healthy pregnant women".

Sri Lanka is a country with an ongoing epidemic of chronic kidney disease of unknown origin (CKDu) [23, 24]. Anuradhapura, where the present study was performed, is one of the most affected districts [25]. A previous study performed in the same study area among pregnant women showed a mean eGFR of 145.5 mL/min/1.73 $m^2$ [26], which is higher than the numbers presented in the present study (122–130 mL/min/1.73 $m^2$). That particular study was not conducted specifically among healthy pregnant women; thus, the eGFR estimates may be slightly different. In the same study area, early renal damage among children was proposed [27], raising the question of whether CKDu is partly due to an early environmental impact. Based on these observations, a higher prevalence of renal problems might be expected even among pregnant women showing high mean sCr values. Nevertheless, the use of the nonpregnant comparison group and application of percentage increase will overcome this issue when generalizing the results.

CKD-EPI was used in this study to estimate the eGFR. While this formula has been shown to underestimate eGFR during pregnancy [28], CKD-EPI has good performance postpartum and outside pregnancy, and the current evidence does not suggest that a superior formula is available for eGFR estimation in pregnancy [29]. Other confounding factors which could influence sCr such as diet exercise were not considered in this analysis because we were not looking at an association.

To strengthen the observations and to evaluate the utility of the proposed normality data, a long follow-up of the same cohort is required with proper assessment of maternal and fetal outcomes. Although the sCr-based eGFR is not an accurate estimate during pregnancy, previous studies have shown that it could be used as a predictor of adverse pregnancy outcomes [9, 10]. As the normality data generated through this study are almost similar to the values observed in the previous secondary data analysis, these values seem universally valid across geographical regions [13].

## Conclusions

This prospective cohort study provides the threshold values for sCr during the first and second trimesters of pregnancy in a South Asian Population. The rapid decrease in early pregnancy sCr and differences across trimesters need to be considered during clinical practice while interpreting sCr in pregnancy. Extension of prospective studies from early pregnancy to late infancy will provide confirmatory data on the upper threshold values for sCr as a biomarker of adverse pregnancy outcomes.

## Supporting information

**S1 Data.**
(XLSX)

**S1 Text.**
(DOCX)

## Acknowledgments

We acknowledge the north-central province and Anuradhapura district public health staff for the support given during this study.

## Author Contributions

**Conceptualization:** Suneth Buddhika Agampodi.

**Data curation:** Suneth Buddhika Agampodi.

**Formal analysis:** Suneth Buddhika Agampodi.

**Funding acquisition:** Suneth Buddhika Agampodi, Thilini Chanchala Agampodi.

**Investigation:** Suneth Buddhika Agampodi, Gayani Shashikala Amarasinghe, Janith Niwanthaka Warnasekara, Ayesh Umeshana Hettiarachchi, Imasha Upulini Jayasinghe, Iresha Sandamali Koralegedara, Parami Abeyrathna, Shalka Srimantha, Farika Nirmani de Silva, Sajaan Praveena Gunarathne, Nuwan Darshana Wickramasinghe.

**Methodology:** Suneth Buddhika Agampodi, Thilini Chanchala Agampodi, Gayani Shashikala Amarasinghe, Janith Niwanthaka Warnasekara, Ayesh Umeshana Hettiarachchi, Imasha Upulini Jayasinghe.

**Project administration:** Suneth Buddhika Agampodi, Thilini Chanchala Agampodi, Ayesh Umeshana Hettiarachchi.

**Resources:** Suneth Buddhika Agampodi, Thilini Chanchala Agampodi.

**Software:** Suneth Buddhika Agampodi.

**Supervision:** Thilini Chanchala Agampodi.

**Validation:** Suneth Buddhika Agampodi.

**Visualization:** Suneth Buddhika Agampodi.

**Writing – original draft:** Suneth Buddhika Agampodi.

**Writing – review & editing:** Thilini Chanchala Agampodi, Gayani Shashikala Amarasinghe, Janith Niwanthaka Warnasekara, Ayesh Umeshana Hettiarachchi, Imasha Upulini Jaya-singhe, Iresha Sandamali Koralegedara, Parami Abeyrathna, Shalka Srimantha, Farika Nir-mani de Silva, Sajaan Praveena Gunarathne, Nuwan Darshana Wickramasinghe.

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
