## [Decision Letter · Decision Letter 0]

1 Jul 2022

PGPH-D-22-00223

Renal Functions in early pregnancy and changes during the pregnancy

Dear Dr. Agampodi,

Thank you for submitting your manuscript to PLOS Global Public Health. After careful consideration, we feel that it has merit but does not fully meet PLOS Global Public Health’s publication criteria as it currently stands. Therefore, we invite you to submit a revised version of the manuscript that addresses the points raised during the review process.

The manuscript has been evaluated by two reviewers, and their comments are available below. They request additional information on methodological aspects of the study, interpretation of the results, and generalizability of the findings. Could you please revise the manuscript to carefully address the concerns raised?

We look forward to receiving your revised manuscript.

Kind regards,

Dario Ummarino, PhD

Senior Editor

Journal Requirements:

1. In the online submission form, you indicated that "The datasets generated during and/or analyzed during the current study are available from the corresponding author on reasonable request." require all data underlying the findings described in their manuscript to be freely available to other researchers, either 1. In a public repository, 2. Within the manuscript itself, or 3. Uploaded as supplementary information.

Additional Editor Comments (if provided):

Reviewers' comments:

Reviewer's Responses to Questions

**Comments to the Author**

1. Does this manuscript meet PLOS Global Public Health’s publication criteria? Is the manuscript technically sound, and do the data support the conclusions? The manuscript must describe methodologically and ethically rigorous research with conclusions that are appropriately drawn based on the data presented.

Reviewer #1: Partly

Reviewer #2: Yes

2. Has the statistical analysis been performed appropriately and rigorously?

Reviewer #1: I don't know

Reviewer #2: I don't know

3. Have the authors made all data underlying the findings in their manuscript fully available (please refer to the Data Availability Statement at the start of the manuscript PDF file)?

Reviewer #1: Yes

Reviewer #2: Yes

4. Is the manuscript presented in an intelligible fashion and written in standard English?

Reviewer #1: Yes

Reviewer #2: Yes

5. Review Comments to the Author

Reviewer #1: These data are very useful for producing a local reference range for serum creatinine in pregnancy and the creatinine data provided should be made available for future meta-analyses. However there are some limitations which restrict generalisability to a global readership:

1. Reported creatinine concentrations are limited to 1st and 2nd trimesters, yet most creatinine concentrations measurements in pregnancy are taken in the third trimester, where AKI is most prevalent due to peripartum events.

2. It is not clear whether the serum creatinine method is an isotope dilution mass spectometry traceable creatinine assay

3. eGFR is reported as a measure of kidney function in pregnancy even though it is acknowledged that it consistently underestimates function. Existing published data on the association of eGFR with adverse pregnancy outcomes utilises eGFR as a biomarker, rather than an absolute measure of kidney function. This is distinct from the establishment of a reference interval of eGFR as a marker of kidney function, which is flawed.

4. Up to one third of CKD presents for the first time in pregnancy. Although a known diagnosis of CKD and the presence of hypertension at the time of the study are exclusion criteria, an opportunity was missed to exclude haematuria and proteinuria, which may have helped to make the data more robust.

5. The value of the data to other centres, especially given the local prevalence of CKDu, is not clear. The data are certainly smaller than that reported by others using a single assay eg JAMA. 2019;321(2):205-207. doi:10.1001/jama.2018.17948, n=243,534.

Reviewer #2: This is a prospective study of serum Cr values in early pregnancy and throughout gestation, up to 30 weeks. The authors define 'normal' ranges for pregnancy in the healthiest population they could identify and compared it to a similar population. This is an important paper and adds solid, clear data to a field with a lot of ambiguity. It is helpful to have this data in the South Asian population, specifically, as well.

I have a few small points:

1. I am a bit confused about women who were included. In the methods, it states 'only women who made it 24 weeks were included', but then there seems to be a separate group studied for the repeated measures analysis. In Table 3, I do not really understand what the N column is referring to - is this just the number of women who had their first Cr drawn at that point in gestation but that whole group definitely had a check at 24 weeks? Some more explanation is needed, I think on this point as to the number of blood draws each woman had and what the different groupings mean.

2. There are a lot of statistical methods described in the results, but not elaborated on in the methods. Would make sure to describe the tests and why they were chosen and used in the methods, as well.

3. Is the effect of age on eGFR just due to the fact that age is in the CKD-EPI equation in a non-linear format? Age is an exponent, so I would anticipate that there would be non-linear effects of age on eGFR in every circumstance. If there is more to it than this, which there very well could be, would explain in the discussion what the meaning of the interaction testing, etc. you did in the results and how people should be thinking about age when determining what values are outside the normal range in pregnancy, if at all.

6. PLOS authors have the option to publish the peer review history of their article (what does this mean?). If published, this will include your full peer review and any attached files.

**Do you want your identity to be public for this peer review?** For information about this choice, including consent withdrawal, please see our Privacy Policy.

Reviewer #1: No

Reviewer #2: No

---

## [Decision Letter · Decision Letter 1]

3 Jan 2023

Serum Creatinine and estimated glomerular filtration rate (eGFR) in early pregnancy and changes during the pregnancy

PGPH-D-22-00223R1

Dear Prof Agampodi,

We are pleased to inform you that your manuscript 'Serum Creatinine and estimated glomerular filtration rate (eGFR) in early pregnancy and changes during the pregnancy' has been provisionally accepted for publication in PLOS Global Public Health.

Best regards,

Julia Robinson

Executive Editor

Reviewer Comments (if any, and for reference):

Reviewer's Responses to Questions

**Comments to the Author**

1. If the authors have adequately addressed your comments raised in a previous round of review and you feel that this manuscript is now acceptable for publication, you may indicate that here to bypass the “Comments to the Author” section, enter your conflict of interest statement in the “Confidential to Editor” section, and submit your "Accept" recommendation.

Reviewer #2: All comments have been addressed

2. Does this manuscript meet PLOS Global Public Health’s publication criteria? Is the manuscript technically sound, and do the data support the conclusions? The manuscript must describe methodologically and ethically rigorous research with conclusions that are appropriately drawn based on the data presented.

Reviewer #2: Yes

3. Has the statistical analysis been performed appropriately and rigorously?

Reviewer #2: Yes

4. Have the authors made all data underlying the findings in their manuscript fully available (please refer to the Data Availability Statement at the start of the manuscript PDF file)?

Reviewer #2: Yes

5. Is the manuscript presented in an intelligible fashion and written in standard English?

Reviewer #2: Yes

6. Review Comments to the Author

Reviewer #2: All of my comments have been addressed.

7. PLOS authors have the option to publish the peer review history of their article (what does this mean?). If published, this will include your full peer review and any attached files.

**Do you want your identity to be public for this peer review?** For information about this choice, including consent withdrawal, please see our Privacy Policy.

Reviewer #2: No
